# OpenReview forum: "ARGS: Alignment as Reward-Guided Search"
_ICLR.cc/2024/Conference — ICLR 2024 poster_

### Official Review · Reviewer_6nE9 · 2023-10-21

**Soundness:** 2 fair
**Presentation:** 1 poor
**Contribution:** 2 fair
**Rating:** 6
**Confidence:** 4

**Summary:**

The paper introduces ARGS, a decoding framework that enhances the alignment of generated text with human preferences. It achieves this by employing a reward mechanism that guides the text-generation process of a language model. The method consists of reward-guided scoring and token selection. The goal is to generate text that is both coherent and contextually relevant while satisfying specific alignment criteria or objectives. The method improves the average reward compared to standard decoding and demonstrates better lexical diversity without compromising contextual consistency. The experiments validate the effectiveness of ARGS in aligning the generated text with human preference.

**Strengths:**

- Authors aim to resolve an important problem in the reasearch area.

- The problem is interesting.

- Good discussion on broader impacts.

**Weaknesses:**

- There are some serious issues in citation. "Decoupled weight decay regularization" is an ICLR-19 paper, not an arxiv. Please refer to https://openreview.net/forum?id=Bkg6RiCqY7. The authors should list all wrong citations and revise them. I will check the similar issues in all citations one by one.

- Qualitative results are limited. I suggest that the authors provide more results to support the claims. It is hard for me to have a clear understanding of the improvements. If an anonymous web demo or a code link is provided, I will revise my rating.

- In Figure 2, different types of lines are not shown in the figure. It is not very clear.

- Compared with classical decoding methods, ARGS has higher time complexity. Although the $k$ can be small, it also has a higher complexity. Besides, small $k$ is not good for performance.

- No period in Equation $T_{ARGS}(n, m, k)$.

- The latest baseline method is a paper published in 2022. More baselines should be compared.

- I am concerned about the technical novelty of the paper. The idea of reward-guided search has been proposed in SCST (Self-Critical Sequence Training). Besides, the technical contribution compared with the baseline is a tricky implementation, which is marginal.

- The user study is needed for evaluation.

- Missing discussion on limitation.

Overall, the writing of this submission is unprofessional and the technical contribution is marginal. I provide a reject rating here and I will revise the rating according to the authors; rebuttal and other reviews.

**Questions:**

See weakness.

---

Revise rating from 3 to 5, 5->6.

---

> ### Author Response · Authors · 2023-11-15
> **Response to Reviewer 6nE9 - Part I**
>
> We sincerely thank the reviewer for the detailed review, which has helped us strengthen the manuscript. Below we address your comments and suggestions:
>
> > **Citation format**
>
> We appreciate your attention to detail. During the manuscript preparation, we made a conscientious effort to adhere to the proceeding format when citing relevant papers. However, we acknowledge the possibility of oversights during the editing phase. As suggested, we conducted a comprehensive check and revised several citations, including the one you highlighted.
>
> It is important to note that some of the cited papers currently exist solely on arXiv without an associated proceeding venue, primarily due to their recent release, especially for works originating in 2023. For these instances, we have retained the citations in their current form and will update them when they are formally published.
>
> > **Qualitative results and code link**
>
> In Section 3.3, we conducted qualitative evaluations across 300 randomly sampled prompts from the test set of HH-RLHF. Table 2 presents the qualitative evaluation results, measured by the percentage of win-ties of our method over the alternative decoding strategies. During rebuttal, we have updated our draft by incorporating more qualitative examples. For details, we direct the reviewer to **Appendix D**.
>
> To facilitate further scrutiny of our work, we have included an anonymous link to our code (in the global response to all reviewers). This link will provide you with access to the experiments and implementation details.
>
> > **Figure 2**
>
> That's a great catch. For clarity, we have revised the figure by changing the legend (ARGS-greedy 10 and 40) to a rectangle instead of a line.
>
> > **Time complexity**
>
> We would like to point out that employing small $k$ yields similar performance as large $k$. This has been empirically validated in **Section 3.2**. Specifically, Figure 2 illustrates that the performance variation between $k=40$ (darker color bars) and $k=10$ (light color bars) is slight, suggesting that a large number of candidates may not be essential for producing aligned generations.
>
> Empirically, we find that the time complexity overhead can be considerably small. For example, using the OPT-2.7b base model, the generation time per response increases only by 1.9 times when comparing ARGS ($k=10$) with conventional greedy decoding. Despite the slight increase in processing time, there was a notable enhancement in reward performance by $\uparrow$ 6.8%, demonstrating the existence of a reasonable tradeoff between reward optimization and inference speed. The gap can be further reduced by employing a smaller reward model, or parallelizing the reward computation across $k$ candidates.
>
> Our discussion on time complexity has also been endorsed by reviewer Mqii, who commented:
> > _"The authors do discuss the extra computation added at inference time and show its feasibility. I believe that **such a method is interesting and useful for the literature even with this extra weight at inference**. For example, it can be used to iterate over different reward models before running only one finetuning, or used directly if we have a small enough and good reward model."_
>
>
> > **More baselines**
>
> We have added a latest baseline DPO (Direct Preference Optimization) [1] from NeurIPS 2023. The results have been added to **Section 4**.
>
> > **Human evaluation**
>
> We employ a GPT-4-based evaluation approach to assess the quality of responses. Research, as discussed in [2], has shown that using a GPT-4 proxy aligns with human evaluations in over 80% of cases, providing a scalable method to approximate human preferences. This evaluation method is prevalent in recent literature, as evidenced by studies such as [1, 3].
>
> Due to practical considerations, our Institutional Review Board (IRB) protocol for conducting human evaluations is currently undergoing review at our institution. We fully intend to incorporate human evaluations into our study as soon as our IRB protocol receives approval.

---

> > ### Author Response · Authors · 2023-11-15
> > **Response to Reviewer 6nE9 - Part II**
> >
> > > **Related work**
> >
> > We thank you for pointing out the literature [4]. Upon close examination, the work appears to be different from ours in terms of problem setting, methodology, and various experimental details.
> >
> > - Problem setting: While SCST focuses on the problem of image captioning, our distinct focus is the alignment of large language models. To the best of our knowledge, we are the first to introduce a reward-guided search mechanism for the alignment problem.
> > - Methodology: SCST does not train the reward model using human preferences like we do, and directly employs greedy decoding in inference time. Instead, our key intellectual contribution is a reward-guided decoding procedure in testing time, which combines explicitly reward scores as well as the output of language models. Moreover, SCST involves an RL algorithm, whereas our framework circumvents the need for RL training.
> > - Other empirical distinctions: SCST was developed based on LSTM and CNN architectures, whereas the alignment literature today commonly focuses on transformer-based autoregressive language models. It's also worth noting that the evaluations, experiments, and metrics are all different between the two works.
> >
> >
> > > **Discussion on limitations**
> >
> > As suggested, such a discussion has been added to our revised manuscript. Please see **Section 6**.
> >
> >
> > We hope our response has helped address your concerns. In light of the changes, we hope the reviewer will consider re-evaluating our work. Thanks again for your valuable time and service.
> >
> > ------
> > References
> >
> > [1] Rafael Rafailov, Archit Sharma, Eric Mitchell, Stefano Ermon, Christopher D. Manning, and Chelsea Finn. Direct preference optimization: Your language model is secretly a reward model. Advances in Neural Information Processing Systems, 2023.
> >
> > [2] Lianmin Zheng, Wei-Lin Chiang, Ying Sheng, Siyuan Zhuang, Zhanghao Wu, Yonghao Zhuang, Zi Lin, Zhuohan Li, Dacheng Li, Eric. P Xing, Hao Zhang, Joseph E. Gonzalez, and Ion Stoica. Judging llm-as-a-judge with mt-bench and chatbot arena. arXiv preprint arXiv:2306.05685, 2023a.
> >
> > [3] Wei-Lin Chiang, Zhuohan Li, Zi Lin, Ying Sheng, Zhanghao Wu, Hao Zhang, Lianmin Zheng, Siyuan Zhuang, Yonghao Zhuang, Joseph E. Gonzalez, Ion Stoica, and Eric P. Xing. Vicuna: An open-source chatbot impressing gpt-4 with 90%* chatgpt quality. https://vicuna.lmsys.org, 2023.
> >
> > [4] Rennie, Steven J., Etienne Marcheret, Youssef Mroueh, Jerret Ross, and Vaibhava Goel. Self-critical sequence training for image captioning. In Proceedings of the IEEE conference on computer vision and pattern recognition, pp. 7008-7024. 2017.

---

> > > ### Comment · Reviewer_6nE9 · 2023-11-16
> > > **RE: Response to Reviewer 6nE9**
> > >
> > > Dear authors,
> > >
> > > Thanks for the authors' reply. Up to now, my concern has been partially resolved. (1) The remaining concern is about the time complexity. I will discuss this with reviewer Mqii. (2) Besides, I suggest discussing SCST in the paper. (3) For the provided codes, I will have a try in the following days.
> > >
> > > I enjoy the process of polishing a paper with authors. If all three can be resolved in the following days, I will revise my rating. And I will also follow other reviews.
> > >
> > > Best,
> > >
> > > Reviewer

---

> > > ### Comment · Reviewer_6nE9 · 2023-11-16
> > > **Reminder**
> > >
> > > The revision seems to have exceeded the space limit, please be attentive. Try to avoid factors that may cause desk rejection.

---

> > > > ### Author Response · Authors · 2023-11-16
> > > > **Response**
> > > >
> > > > Thank you for your quick response! We are glad to hear that some of your concerns have been resolved. As suggested, we have also revised our paper to include a discussion on SCST [1]. Please see Section 5.
> > > >
> > > > We also appreciate your reminder on the space limit. We strictly followed the ICLR guideline (https://iclr.cc/Conferences/2024/AuthorGuide) to ensure compliance with the policy. In particular, the ethics statement is allowed beyond the 9 page limit.
> > > >
> > > > > _"Authors are encouraged to include a paragraph of Ethics Statement (at the end of the main text before references).  The optional ethic statement will not count toward the page limit, but should not be more than 1 page."_
> > > >
> > > > [1] Rennie, Steven J., Etienne Marcheret, Youssef Mroueh, Jerret Ross, and Vaibhava Goel. Self-critical sequence training for image captioning. In Proceedings of the IEEE conference on computer vision and pattern recognition, pp. 7008-7024. 2017.

---

> > > > > ### Comment · Reviewer_6nE9 · 2023-11-17
> > > > > **RE: Response**
> > > > >
> > > > > Dear authors,
> > > > >
> > > > > Thanks for the clarification on space limits. Accodring to review guidance, my discussion with reviewer Mqii will be after the author-reviewer discussion phase. However, I will run your codes to decide whether to revise my rating. If I do not reply before Nov. 20th 23:59 (AoE), please write an official comment to remind me. If you have any concerns, feel free to tell me. Thanks.
> > > > >
> > > > > Best,
> > > > >
> > > > > Reviewer

---

> ### Author Response · Authors · 2023-11-18
> **Additional resources: Model checkpoints**
>
> We thank you for the follow-up and for engaging actively in the discussion!
>
> In case it is helpful, we have also included the checkpoints of SFT and reward models. This allows you to perform decoding based on our approach directly. Please check them out in the revised README.
>
> Best,
>
> Authors

---

> > ### Comment · Reviewer_6nE9 · 2023-11-18
> > **RE: Additional resources: Model checkpoints**
> >
> > Thanks for the more details. It may help me to check the codes and conduct testing. I will do it ASAP.
> >
> > Best,
> >
> > Reviewer

---

> > > ### Comment · Reviewer_6nE9 · 2023-11-21
> > > **Revising rating**
> > >
> > > Dear authors,
> > >
> > > Thanks for providing the codes. I have checked the codes you provided. I would like to revise my rating from 3 to 5 now. My remaining concern is only about the time complexity. I will discuss this with reviewer Mqii in the next session. This determines whether I will continue to increase the rating. Besides, will you provide the web demo if accepted? If yes, I will increase my rating, too.
> > >
> > > Best,
> > >
> > > Reviewer

---

> > > > ### Author Response · Authors · 2023-11-21
> > > > **RE: Revising rating**
> > > >
> > > > Thank you for taking the time to check our provided codebase and increasing the score!
> > > >
> > > > In response to your query about a web demo, we are indeed planning to provide one, likely through Huggingface space, upon acceptance. This will facilitate public testing and engagement with our work!
> > > >
> > > > Best,
> > > >
> > > > Authors

---

> > > > > ### Comment · Reviewer_6nE9 · 2023-11-22
> > > > > **RE: RE: Revising rating**
> > > > >
> > > > > I revise my rating from 5 to 6. Authors should do as promised.

---

> > > > > > ### Author Response · Authors · 2023-11-22
> > > > > > **thank you**
> > > > > >
> > > > > > Thank you - we really appreciate it!

---

### Official Review · Reviewer_Mqii · 2023-10-30

**Soundness:** 3 good
**Presentation:** 3 good
**Contribution:** 3 good
**Rating:** 8
**Confidence:** 3

**Summary:**

This paper introduces ARGS, a new framework for aligning LLMs with human preferences without the expensive RL training (i.e., RLHF). To this end, ARGS aligns the LLM with human preferences during the decoding step. Through a set of experiments, the authors show that ARGS leads to better alignment and diversity than the non-aligned baselines while preserving good coherence.

**Strengths:**

This paper introduces ARGS; a simple decoding-based model for LLMs alignment with reward models. In particular, ARGS only introduces an additional hyper-parameter "w" to tune at inference time.

This method is simple and leads to competitive results compared to PPO while not requiring any finetuning step. The authors do discuss the extra computation added at inference time and show its feasibility. I believe that such a method is interesting and useful for the literature even with this extra weight at inference. For example, it can be used to iterate over different reward models before running only one finetuning, or used directly if we have a small enough and good reward model.

**Weaknesses:**

This paper aims to replace the RL step for human alignment with a more lightweight, only decoding-based, process. This means that PPO (as noted in Table 4) is the main baseline for ARGS. However, this comparison is not elaborated enough in the paper. This work focuses instead on other decoding-based baselines that do not aim for human alignment. For example, it would be interesting to add the Win-Tie(%) results for ARGS vs. PPO in Table 2 and discuss the low "Diversity" numbers for PPO in Table 4 (is it an issue of the KL penalty, was it hard to cross-validate this term?) These points would be an interesting addition to this paper.

**Questions:**

.

---

> ### Author Response · Authors · 2023-11-15
> **Response to Reviewer Mqii**
>
> We sincerely thank the reviewer for the positive and constructive review. We are very grateful that you find our paper useful for the literature. We address your feedback below!
>
> > **GPT-4 Evaluation of ARGS and baseline PPO**
>
> Following your suggestion, we have conducted an additional GPT-4 evaluation, comparing ARGS with the baseline PPO (on OPT models from Section 4). This evaluation strictly follows the protocol detailed in Section 3.3. The table below illustrates the completed GPT-4 evaluation results, measured by the percentage of win-ties of our method. Overall our method ARGS produces more favorable responses, achieving a win-tie rate of 72.33%.
>
> |  | Method | Win-Tie (%) |
> | ---- | --------------- | ---------------------- |
> | ARGS | PPO             | 72.33                  |
>
> > **Low diversity for PPO**
>
> That's a great point. In the table below, we include the results of vanilla greedy decoding on the finetuned OPT-1.3b (SFT) which was the base model for the PPO model to validate the low diversity of PPO. The relatively low diversity of the outputs produced by the PPO model can be potentially related to the low diversity of the SFT model, as a result of the KL penalty.
>
> | Method      | Average Reward | Diversity | Coherence |
> | ----------- | -------------- | --------- | --------- |
> | ARGS-Greedy | 5.98           | 0.322     | 0.390     |
> | PPO-Greedy  | 5.88           | 0.247     | 0.264     |
> | SFT-Greedy  | 5.35           | 0.236     | 0.280     |

---

### Official Review · Reviewer_2sRW · 2023-10-31

**Soundness:** 3 good
**Presentation:** 3 good
**Contribution:** 3 good
**Rating:** 6
**Confidence:** 4

**Summary:**

The paper introduces a novel framework called ARGS (Alignment with Reward-Guided Sampling) for aligning language models with human preferences. The framework offers a flexible and efficient solution that eliminates the need for expensive RL training. With ARGS, you can generate texts with semantic diversity while being aligned with human objectives.

**Strengths:**

1. Resource-efficient: The ARGS framework is designed to be resource-efficient, making it an ideal solution for smaller institutions or businesses without the capacity for large-scale training. This can potentially level the playing field, allowing those with limited computational resources to benefit from state-of-the-art models without incurring significant costs.

2. Broader applicability: The compatibility of the ARGS framework with different reward models extends its applicability across various domains and industries. This can accelerate the adoption of machine learning solutions in fields where resource constraints or rapidly changing data are prevalent.

3. Easy to integrate: The ARGS framework is easy to integrate into existing language models, making it a practical solution for aligning language models with human preferences. The authors provide a detailed explanation of how to integrate ARGS into a pre-trained GPT-3 model, making it accessible to a wider range of users.

**Weaknesses:**

1. Limited evaluation: The evaluation of the ARGS framework is limited to a few specific tasks (e.g., harmfulness), and it is unclear how well the framework would perform on other tasks (more complex ones like multi-step reasoning), especially when a good reward model is not easy to train. This may limit its applicability in certain domains.

2. Unfair evaluation: The evaluation of the ARGS framework is evaluated on the score from reward model. However, there are some limitations: (1) ARGS will certainly achieve higher scores since the RM is integrated during the decoding process. Essentially, applying any RM constraints during the decoding stage will result in higher scores when being evaluated by that RM. (2) The calibration of RM remains unclear -- does a higher reward score certainly lead to a better response, especially the reward score difference is less than 1?

Overall, I think it not very fair if the authors use the same RM in their methods and evaluation.

**Questions:**

1. A good reward model is vital in ARGS, have you ever tried to enhance the RM? The paper does not show the relation between **RM quality** and **Effectiveness of ARGS**. For example, you might use some techniques from [a,b,c] to strengthen your RM.

(a) The Trickle-down Impact of Reward (In-)consistency on RLHF 2023

(b) Aligning Language Models with Preferences through f-divergence Minimization 2023

(c) Fine-Grained Human Feedback Gives Better Rewards for Language Model Training 2023

2. How would you evaluate the instruction-following ability of model based on ARGS? Since Diversity/Coherence are measuring the naturalness of model responses, there is a blank space for quantifying whether model completes the instruction (in your case, it is ethical property).

---

> ### Author Response · Authors · 2023-11-15
> **Response to Reviewer 2sRW - Part I**
>
> We sincerely thank the reviewer for the detailed and constructive review. We are also grateful that you find our paper novel and useful for the broader community. We address your feedback below!
>
> > **Evaluation on other tasks**
>
> You raised an excellent point. For our current evaluations, we follow the standard and commonly used benchmarks in alignment literature. In particular, HH-RLHF from Anthropic and Stanford Human Preferences (SHP) are among the largest and publicly available datasets for alignment research. These tasks allow us to draw comparisons with existing approaches more easily and directly. As the first study to introduce the decoding-time alignment approach, we believe it's important to understand the potential of our approach more deeply in the existing realm of alignment tasks, before we expand to other tasks.
>
> With that being said, we do agree that evaluating on more complex tasks such as multi-step reasoning would be valuable. During rebuttal, we looked further into this, however, we could not locate large-scale human preference datasets on multi-step reasoning, which hinders the feasibility of training the reward model. We have revised our draft to acknowledge this limitation and remain interested in delving deeper into more complex tasks as part of our future work.
>
> > **Evaluation metrics**
>
> We understand your concern out of the Goodhart's Law :) We share the same and thus have taken the following additional steps to ensure our evaluations are comprehensive:
>
> - We performed an evaluation where the evaluating reward model is _different_ from the one we used in training. In Section 3.4, we conducted experiments where we employ fined-tuned OPT-1.3b and OPT-2.7b for decoding, and OPT-125m as reward models. We evaluate the models on random 1,000 samples of the test set from the Stanford Human Preferences (SHP) dataset, and the average reward is calculated by the OPT-350m reward model. As shown in Figure 3, ARGS consistently outperforms the greedy baseline.
> - To address the nuanced aspects of language quality that the standard metrics (diversity, coherence, reward) may not comprehensively capture, we also adopt a GPT-4-based evaluation approach for comparing the quality of responses. Table 2 presents the GPT-4 evaluation results, measured by the percentage of win-ties of our method over the alternative decoding strategies. A higher percentage indicates that our proposed method is more proficient in generating responses that exhibit not only contextual relevance and accuracy but also helpfulness and harmlessness. This observation is consistent with the outcomes of the reward-based evaluation discussed in Section 3.2.
>
> > **Relation between RM quality and effectiveness of ARGS**
>
> Another great point raised!
>
> Indeed, our experimental results in Section 3.4 reveal such a connection between RM quality and the effectiveness of ARGS. In particular, the RM quality can be modulated by the model capacity. We experimented with a smaller capacity model OPT-125M, along with a larger capacity model OPT-350M. The reward modeling accuracy, as well as the ARGS performance (average reward, diversity, coherence), is summarized in the table below. We observe that a higher RM accuracy in general leads to stronger decoding performance by ARGS.
>
> **Base model**: OPT-1.3b
>
> | Reward Model | Evaluation Accuracy (%) | Average Reward | Diversity | Coherence |
> | ------------ | ----------------------- | -------------- | --------- | --------- |
> | OPT-125m     | 52.62                   | 5.698          | **0.322** | 0.376     |
> | OPT-350m     | **53.16**               | **5.979**      | **0.322** | **0.389** |
>
> **Base model**: OPT-2.7b
>
> | Reward Model | Evaluation Accuracy (%) | Average Reward | Diversity | Coherence |
> | ------------ | ----------------------- | -------------- | --------- | --------- |
> | OPT-125m     | 52.62                   | 5.71           | 0.369     | 0.431     |
> | OPT-350m     | **53.16**               | **5.929**      | **0.380** | **0.435** |
>
> This experiment also informs us about a meaningful future direction to explore ARGS decoding from an RM modeling perspective. The papers you recommended are excellent starting points for this. We hypothesize that coupling advanced RM modeling objectives (beyond pairwise ranking loss) with ARGS can help further enhance the generation quality. We have added those discussions in our updated manuscript accordingly.

---

> > ### Author Response · Authors · 2023-11-15
> > **Response to Reviewer 2sRW - Part II**
> >
> > > **Instruction following evaluation**
> >
> > We evaluate the instruction following ability based on GPT-4 evaluation (Section 3.3). We explicitly instruct GPT-4 to assign the score to the responses based on helpfulness, harmlessness, relevance, accuracy, and insightfulness (see the prompt template attached in **Appendix B**).
> >
> > Table 2 presents the GPT-4 evaluation results, measured by the percentage of win-ties of our method over the alternative decoding strategies. A higher percentage indicates that our proposed method is more proficient in generating responses that exhibit not only contextual relevance and accuracy but also helpfulness and harmlessness.

---

> > ### Comment · Reviewer_2sRW · 2023-11-21
> >
> > Hi! Thanks for your response! It resolves most of my concerns. Still, I feel that only scaling up the RM model size is incremental to enhance the capability of RM. It would be interesting to see what will happen IN ARGS if advanced techniques are applied to strengthen the RM or if the RM has some obvious issues like inconsistency [a].
> >
> > Overall, I believe the authors have resolved most of my concerns. Thus, I decide to increase the score to 6.
> >
> >
> > (a) The Trickle-down Impact of Reward (In-)consistency on RLHF 2023

---

> > > ### Author Response · Authors · 2023-11-21
> > > **RE: Official Comment by Reviewer 2sRW**
> > >
> > > Thank you for taking the time to read our response and increasing the score! We remain very interested in exploring coupling ARGS with advanced reward modeling techniques, which may be worth a full-scope study on its own. We believe our work provides an important foundation to enable these exciting possibilities in future work.

---

### Official Review · Reviewer_vytn · 2023-11-01

**Soundness:** 3 good
**Presentation:** 3 good
**Contribution:** 3 good
**Rating:** 8
**Confidence:** 3

**Summary:**

This paper addresses the question: Do we really have to only have one model for sampling language? Recent work distills or amortizes reward models into language models via PPO or DPO, so that sampling is simple. This paper proposes a method for avoiding the distillation step. They instead perform decoding with the product of experts obtained by combining a language model with a reward model. The method does not require training; it directly uses the reward model, trained only on scoring complete sequences, to score incomplete prefixes during search.

Experiments show that, compared to the SFT baseline, taking a product of experts ensemble of the language and reward models on incomplete prefixes during search results in better average rewards overall.

**Strengths:**

There are two steps in this paper:
1. Decoupling reward and language models as a product of experts (PoE)
2. Using the reward model, unmodified, on prefixes

The first idea is the primary focus of the paper, and the second idea is not discussed. The second idea is just as, if not more important than the first. The reason PPO or DPO is used in the first place is that the reward model is an energy-based model that scores complete sequences, which can be amortized into a left-to-right autoregressive policy. This work bypasses that issue by directly applying the reward model to incomplete prefixes. The application of the reward model to prefixes instead of complete sequences requires experimental justification -- more on this in the weaknesses.

Other than that, the originality, clarity, and significance were good.

**Weaknesses:**

The decision to directly apply the reward model on prefixes should be justified experimentally, and separately from the decision to decouple the language and reward models. The main question I am interested in is: What is the performance loss from using the reward function on incomplete prefixes? Secondly, when is the predicted reward from the reward function most unreliable (likely on sequences further from completion)?

Separately, I understand that DPO [1] could maybe be considered concurrent work, but there should be comparisons against it.

[1] Rafael Rafailov, Archit Sharma, Eric Mitchell, Stefano Ermon, Christopher D Manning, and Chelsea Finn. Direct preference optimization: Your language model is secretly a reward model. May 2023

**Questions:**

## Questions and comments
1. Can you add a sentence in section 3.1 stating that SFT on HH-RLHF means fine-tuning on the winning responses, if that is what was done.
2. In section 3.2, can you say a relative improvement of 19.56% *in average rewards*.
3. Is PPO the most widely used training-time alignment approach, or DPO?
4. The contributions I would like to see are 1. Decouple reward and LM as product of experts, 2. Show that reward models can be reasonably applied on prefixes, and 3. Experimental validation.

## Experimental ideas to strengthen the paper
1. You could combine multiple reward functions pretty easily.
2. It would be interesting to see if other reward models return sensible rewards on prefixes.

---

> ### Author Response · Authors · 2023-11-15
>
> We sincerely thank the reviewer for the positive and constructive feedback! We are encouraged that you recognize the originality, clarity, and significance of the work. We address your comments below in detail.
>
> > **Further analysis of using reward model on prefixes**
>
> You raise a very insightful question. To better understand this, we analyzed the average reward w.r.t. different locations. Specifically, for a given $\texttt{prompt}$, we denote the sequence of generated tokens as $x_1$, $x_2$,.., $x_t$, and so on. The average reward at $t$-th position is calculated as $r([\texttt{prompt},x_{\le t}])$, averaged over the entire test set of HH-RLHF. We use the SFT-ed Llama-7B model for decoding, and report the statistics below for your reference.
>
> | Predicted token index $t$ | Average Reward |
> | ------------------------- | -------------- |
> | 10th                      | 5.38           |
> | 50th                      | 6.42           |
> | 100th                     | 6.80           |
>
>
> As ARGS predicts more tokens, it can be seen that the average reward monotonically increases. This indicates that using the reward model on partial prefixes can steer the search toward generating sequences with higher overall reward scores.
>
>
>
> > **Comparison with DPO**
>
> As suggested, we additionally conduct comparisons with the latest approach DPO [1] and report the result in the table below. The comparison and full training configurations have also been added to our manuscript (see **Section 4** and **Table 8**). Our method overall achieves a higher average reward. During experimentation, we notice the tendency for DPO to generate sometimes repetitive words, which potentially leads to low diversity. In contrast, our method more stably generates diverse responses.
>
> | Method | Category       | Average Reward | Diversity | Coherence |
> | ------ | -------------- | -------------- | --------- | --------- |
> | ARGS   | Decoding-based | 5.98           | 0.322     | 0.390     |
> | PPO    | Training-based | 5.88           | 0.247     | 0.264     |
> | DPO    | Training-based | 5.65 | 0.096 | 0.372  |
>
> [1] Rafailov, Rafael, Archit Sharma, Eric Mitchell, Stefano Ermon, Christopher D. Manning, and Chelsea Finn. Direct preference optimization: Your language model is secretly a reward model. NeurIPS 2023.
>
> > **Combining multiple reward functions**
>
> That's a really interesting suggestion. As you recognized, our approach indeed offers the flexibility to integrate multiple reward signals in decoding-time, without having to re-train the PPO model. In our current exploration, we focus on using a single reward function, which allows us to draw comparisons with existing approaches more easily and directly. As the first study to introduce a decoding-time alignment approach, we believe it's important to understand the potential of our approach more deeply in the existing realm of alignment tasks, before we expand to more complex settings.
>
> With that being said, we do agree that evaluating more complex tasks, such as multi-step reasoning, would be valuable. To do so, one needs more thoughtful and meaningful construction of the tasks, as well as the availability of diverse human preference datasets to train different reward functions. For this reason, we would like to spend more time diving deeper into this aspect in our future work, and hopefully present more conclusive findings after.
>
> > **Writing suggestions**
>
> All fixed - the changes have been marked in red in our updated manuscript. Thank you for the careful read!

---

> ### Comment · Reviewer_vytn · 2023-11-16
>
> I have updated my score from 6 to 8.
>
> For a future version, could you also report the accuracies of the reward model on classifying the winner given prefixes of various lengths vs the full sequence on HH-RLHF? This would be a nice appendix experiment.

---

> > ### Author Response · Authors · 2023-11-17
> > **thank you**
> >
> > Thank you for taking the time to read our response and increasing the score! We agree the suggested experiment could be intriguing, and will include this in our future version.

---

### Author Response · Authors · 2023-11-15
**Response and revision summary**

We would like to thank all the reviewers for their valuable efforts and helpful comments. We find it particularly encouraging to see that reviewers find that ARGS framework **novel** and **significant** (vytn, 2sRW, Mqii), offering a **useful**, **flexible** and **efficient** alternative to alignment techniques (vytn, Mqii) while remaining **simple** and **computational feasible** (2sRW, Mqii). We are also glad that reviewers recognize our **clear** presentation, **competitive** results and broader impacts (vytn, 2sRW, Mqii, 6nE9).

We respond to each reviewer's comments in detail below. We have also revised the manuscript according to the reviewers' suggestions, with changes marked in red. For your convenience, we summarize the major changes below:

- Added comparison to the latest/concurrent method Direct Preference Optimization (DPO) from NeurIPS 2023, where ARGS's strong performance holds.
- Added a code repository to facilitate reproducibility and broad utility for the community (https://anonymous.4open.science/r/args-6EDB), which includes detailed step-by-step documentation of how to run and evaluate our method.
- Added GPT-4 evaluation between ARGS and PPO.
- Added more qualitative examples in Appendix D.
- Fixed typos, citation issues, and Figure 2.
- Added a discussion on limitations in Section 6.

We believe these changes have helped strengthen our manuscript. Once again, we express our gratitude for your thoughtful and thorough evaluations.

Sincerely,

Authors

---

### Comment · Area_Chair_XFnq · 2023-12-03
**Comparison to BoN and missing discussion of related work**

Dear Authors,

- You are missing decoding-based baselines that use the reward function at inference, a popular approach being Best-of-N where you decode N samples from your language model and pick the highest ranked one. Ideally, this comparison should be done using the same amount of inference compute for ARGS and BoN and performance should be compared as we vary the amount of inference compute.

- I didn't see any mention of guided decoding approaches, which are well-known in NLP.  Some relevant places to see the related works: A concurrent work which proposes a similar idea to ARGS is [Reward Augmented Decoding](https://arxiv.org/abs/2310.09520). Also, check out the related work section about *guided decoding* in this concurrent submission: https://openreview.net/pdf?id=QaODpeRaOK. Can you provide a detailed discussion of how ARGS fit into this current literature? Generally, I think the claims about novelty of ARGS should be toned down.

---

### Meta-Review · Area_Chair_XFnq · 2023-12-04

**Metareview:**

Summary & Strengths: The paper presents ARGS, an alignment approach that uses the reward model on partial generated prefixes to guide the autoregressive decoding process. ARGS's main strengths lie in its simplicity as well as decoding-based approach to LLM alignment, which presents an alternative to fine-tuning based methods. Experiments on two common alignment tasks demonstrate ARGS efficacy over PPO, DPO as well as Best-of-N.

What might be missing / Weaknesses : Several weaknesses were raised during discussion where some of them were only partially addressed, including:
- Insufficient experimental justification for applying the reward model on incomplete prefixes. While authors added an experiment showing monotonic reward improvements for longer prefix, it is unclear whether rewards trained with pairwise human data actually generalize to partial prefixes. This also relates to the evaluation methodology.

- Evaluation methodology: The evaluation methodology could potentially lead to biased results due to the use of the same reward model for both decoding and evaluation, which leads to doubts about reward hacking due to use of reward functions on partial prefixes (which is not done by other alignment approaches).  The [reward overoptimization](https://arxiv.org/abs/2210.10760) paper might be a useful reference to improve evaluation.

- Overclaiming novelty: The authors need to reposition their work with respect to prior literature, which the current revision does not do well. In discussion with the AC, the authors promised to revise their paper to address this concern.

**Justification For Why Not Higher Score:**

See the weakness pointed above. I hope the authors would revise their camera ready version to address these weaknesses.

**Justification For Why Not Lower Score:**

Good paper with a simple idea where all reviewers are in favor of acceptance.

---

### Decision · Program_Chairs · 2024-01-16

Accept (poster)